# Motives and Barriers Related to Physical Activity and Sport across Social Backgrounds: Implications for Health Promotion

**DOI:** 10.3390/ijerph18115810

**Published:** 2021-05-28

**Authors:** Marlene Rosager Lund Pedersen, Anne Faber Hansen, Karsten Elmose-Østerlund

**Affiliations:** 1Centre for Sports, Health and Civil Society, Department of Sports Science and Clinical Biomechanics, Faculty of Health Sciences, University of Southern Denmark, Campusvej 55, 5230 Odense M, Denmark; kosterlund@health.sdu.dk; 2Department of Research and Analysis, University Library of Southern Denmark, 5230 Odense M, Denmark; annefaber@bib.sdu.dk

**Keywords:** motives, motivation, barriers, physical activity, sport, health promotion, public health, social background, literature review, scoping review

## Abstract

Studies have found physical inactivity to be a significant health risk factor and have demonstrated how physical inactivity behaviour varies according to social background. As a result, differences according to social background must be considered when examining motives and barriers related to physical activity and sport. This scoping review examines motives and barriers related to physical activity and sport among people with different social backgrounds, including age, socioeconomic status, gender, ethnic minority background and disability status. A systematic literature search was performed in four scientific databases and yielded 2935 articles of which 58 articles met the inclusion criteria. We identified common motives for physical activity and sport as health benefits, well-being, enjoyment, social interaction, and social support; common barriers as time restrictions, fatigue and lack of energy, financial restrictions, health-related restrictions, low motivation, and shortage of facilities. We also identified several motives and barriers that were specific to or more pronounced among people with different social backgrounds. The knowledge about motives and barriers related to physical activity and sport provided in this article can inform health promotion initiatives that seek to improve public health both in general and when specifically targeting groups of people with different social backgrounds.

## 1. Introduction

Studies have found physical inactivity to be a significant health risk factor, and a decrease in physical inactivity behaviour has been predicted to significantly improve population health and increase life expectancy [1,2]. There is strong evidence for the positive effects of physical activity and sport participation on various physical health parameters [3,4,5] as well as mental health parameters [6,7]. Thus, there is ample evidence to support the need for health promotion initiatives to focus on physical activity and sport as useful instruments in initiatives that seek to improve public health.

The socio-ecological model originally developed by Urie Bronfenbrenner [8] and in turn refined for use in context to health behaviour by Jim Sallis and colleagues [9,10] illustrates that the promotion of participation in physical activity and sport is a complex undertaking. Factors at different levels affect the health behaviour of individuals. Five levels at various distances from the individual are addressed in the model, including the intrapersonal, interpersonal, organisational, community and public policy level [9,10].

The same complexity is present in the theoretical model that Stef Kremers and colleagues introduced in the field of behavioural nutrition and physical activity as a dual-process view on the environment–behaviour relationship [11]. Parallel to Sallis and colleagues, they argue that energy balance-related behaviour cannot be fully understood by searching for determinants in the environment of an individual. They highlight the important role of both moderators and cognitive mediators in understanding the health behaviour of individuals.

Among the prominent moderators regarding health behaviour are social background characteristics, such as age, socioeconomic status, gender, ethnic minority background and disability status. Similarly, prominent cognitive mediators relate to the motivation, motives and barriers of the individual [11].

In previous literature, the built environment has been shown to be influential for the participation in physical activity and sport for different people with different social backgrounds [12,13]. The centrality of social background as well as motivation, motives and barriers when seeking to explain differences in physical activity and sports participation is also well-documented. Several studies as well as some literature reviews have identified significant correlations between various social background characteristics and the participation in physical activity and sport [14,15,16,17,18,19,20,21]. Similarly, a number of studies have identified significant correlations between motivation, motives and barriers on the one hand and participation in physical activity and sport on the other [22,23,24,25].

Most of the studies cited above are cross-sectional and examine how participation in physical activity and sport varies either between individuals with different social backgrounds or according to differences in motivation, motives and barriers. Less common are studies that specifically examine how motivation, motives and barriers related to physical activity and sport variate between people with different social backgrounds. Nevertheless, there is a need to gather and condense the available information in order to provide an overview of how motivation, motives and barriers related to physical activity and sports participation differ between people with different social backgrounds.

Against this background, it is the purpose of our literature review to provide knowledge about similarities and differences in motivation, motives, and barriers within population groups with different social background characteristics. This knowledge can inform health promotion initiatives by allowing for targeted efforts designed for different population groups, which can—ultimately—be helpful in the battle against social inequalities in physical activity and sport participation.

It should be noted that this literature review has been conducted in context to the ongoing research project ‘Moving Denmark’ that examines physical activity and sport participation among the adult Danish population using surveys, interview studies and accelerometer measurements. The research project focuses on the role of motivation, the built environment and social background for physical activity and sport participation. The literature review reported here is one of three conducted in the initial phase of the research project to qualify the data collection in the project. Thus, the research project does include the built environment as an important factor regarding the participation in physical activity and sport, but this aspect is not included in the literature review presented here. Furthermore, since the purpose of the literature review was to qualify the data collection in the research project, a lack of relevance of the identified articles to the Danish context was used as an exclusion criterium in this literature review.

The aim of this scoping review is to examine motivation, motives and barriers related to participation in physical activity and sport among people with different social backgrounds. In terms of social background, we include five central aspects, namely age, socioeconomic status, gender, ethnic minority background and disability status.

## 2. Materials and Methods

This study was conducted as a scoping review, due to the broad nature of the research question, and due to an intention to identify a broad variety of study concepts [26]. The scoping review approach of Peters et al. [27] was used to conduct this review. Preliminary searches on the main issues of the research question were performed in several databases, and the databases with most search results covering both motivational factors, physical activity as well as social background factors were chosen for the final search. The final search strategy for the four databases was developed as a collaboration between two researchers and two research librarians. The final and systematic search strategy evaluating motivation and motives and barriers related to physical activity and sport across social backgrounds was followed by mapping the characteristics of identified studies, including study population, study design, variables, and key findings. A formal synthesis of evidence was not undertaken.

### 2.1. Terminology of the Main Terms (Physical Activity, Sport, Motivation, Motives, and Barriers)

Physical activity can be defined as “any bodily movement produced by skeletal muscle that results in energy expenditure” [28], while sport involves activities having formally recorded histories and traditions, stressing physical exertion through competition within limits set in explicit and formal rules governing role and position relationships [29]. As ‘sport’ can be viewed as a sub-concept of ‘physical activity’, ‘physical activity’ is used as a unified concept throughout this review.

Motivation can be defined as the hypothetical construct used to describe the internal and/or external forces that produce the initiation, direction, intensity, and persistence of behaviour [30]. While motivation is used to describe the process that takes place when the person has to make a decision to take a certain action, the term ‘motive’ is used to describe the specific rationale for performing a particular action [31], and the term ‘barrier’ is used to describe the specific factor that hinders or inhibits a particular action [32]. Despite the difference in the meaning of ‘motive’ and ‘motivation’, ‘motive’ is for practical reasons used as a unified concept throughout this review.

### 2.2. Identifying Relevant Studies

The final literature search was performed in Global Health, Scopus, Sociological Abstracts and SPORTDiscus. Studies published until 1 November 2019 with full text available in English, German, Danish, Swedish, or Norwegian were included. There was no restriction on publication year. Duplicates were removed before review.

### 2.3. Search Strategy

The search strategy was performed by the use of the Boolean operators “AND” and “OR”. Proximity operators were used (W/2, N2 or NEAR/2) to allow up to two words between search terms as well as indifferent word order. Truncations (*) were used, where appropriate, to include as many forms of the search words as possible. Finally, the search terms were a mix of “free text words” and defined keywords (shown as “DE” in the following example from SPORTDiscus/Global Health and as “MAINSUBJECT.EXACT” in Sociological Abstracts). Here is an example of the used search terms: ((Sport* W/2 participati*) OR (Physical W/2 activit*)) AND (Barrier* OR (Self W/2 determination W/2 theor*)) AND ((Education* W/2 background) OR (Ethnic W/2 minorit*) OR Handicap*). The search strategies were aligned between databases as much as discrepancies in defined keywords and syntax allowed. The full search strategy is listed in Table 1.

### 2.4. In- and Exclusion Criteria

Articles were included if they met three criteria: (a) focus on physical activities in a broad understanding (e.g., walking, jogging etc.); (b) focus on motivation, motives and barriers in a broad understanding; (c) focus on social background characteristics (age, socioeconomic status, gender, ethnic minority background and disability status). Furthermore, we included studies from Europe, Oceania, and North America to maximize transferability to Danish culture and norms.

Articles were excluded if physical activities aimed at narrow activity types (e.g., hang-gliding or parkour); if concerning narrow groups of handicaps (e.g., wheel-chair users or people with acquired brain injury); or if concerning narrow ethnic minority groups in a Danish context (e.g., African Americans or Romas).

### 2.5. Study Selection

A total of 3759 references were identified in the four databases. After removing duplicates, 2935 went through title and abstract review, whereof 2727 were irrelevant. All three authors screened 50 articles together to internally validate the screening process, and subsequently they screened a third of the remaining 2727 references each. In case of disagreement, consensus was achieved between all three reviewers for full-text review. All in all, 208 studies were assessed for eligibility, of which 150 were excluded, with reasons. 58 articles were included for analysis and reviewed by two authors (see Figure 1).

### 2.6. Data Extraction

For identification of common themes, a spreadsheet was developed to extract data from all articles included for review. Data extracted included year of publication, study location, study design and method, study population, aim and variable categories as well as main results and conclusions (Appendix A). As data analysis progressed, each article was coded into these categories, first (a) social background, including age, socioeconomic status, gender, people with ethnic minority background and people with disabilities; and secondly (b) motives and barriers.

## 3. Results

Overall, a total of 58 studies dealt with the link between motives and barriers related to physical activity and social background. Further, 26 studies were from Europe, followed by Oceania and North America (14 and 12 studies, respectively). A total of six studies included more than one country (see Table 2). The earliest studies were published in 1981, but most studies (45) were from 2000 and onwards. The majority of studies were cross-sectional and included quantitative and qualitative studies (e.g., survey and interview studies). Three studies were intervention studies [33,34,35] (Appendix A).

A total 38 of the included studies examined motives in relation to physical activity, while 33 studies examined barriers. The total number exceeds 58 because some studies studied both motives and barriers related to physical activity.

With regard to social background, the included studies can be subdivided into five categories according to their main foci: age, socioeconomic status, gender, people with ethnic minority background and people with disabilities. Table 3 shows the number of studies examining each category.

The motives identified in our literature review can be categorised in 13 general themes, including health benefits, physical appearance, enjoyment, social interaction, social support, competition, guided activities, increased body awareness, environments, intrinsic motivation, self-efficacy, readiness to change and well-being. Similarly, 13 general themes can be applied to the barriers identified, including time restrictions, shortage of facilities, lack of knowledge of facilities, lack of social interactions, bad access conditions, shortage of suitable activities, shortage of proper guidance, health-related restrictions, financial restrictions, low motivation, fatigue and lack of energy, uncertain future as well as religious and cultural norms. These categories of motives and barriers were all present among one or more of the social groups examined. Below, we provide an overview of the presence of these categories of motives and barriers among the five social groups examined, and we elaborate on which aspects within these categories that were identified as relevant for the social group in question. At the end of the results section, a table overview of the five groups’ motives and barriers related to physical activity is presented.

### 3.1. Age

A total 31 studies examined age in relation to motives and barriers related to physical activity (Table 3). In this section, we have divided age into three groups: young adults (16 to approximately 40 years), middle-aged people (approximately 40 to 60 years) and elderly people (approximately 60+ years).

Prominent motives for physical activity among young adults were: enjoyment (operationalised as not being bored and because it was fun) [36,37,38,39,40], physical appearance (to improve physical image and physical identity) [38,39,40,41,42] and social interaction (physical activities may create social unity and an arena for meeting new people) [39,42].

Several studies referred barriers to physical activity for young adults as a shortage of facilities and a shortage of suitable activities (targeted at or suitable for young adults) [37,38]. Another barrier among young adults was time restrictions operationalised as general lack of time [36,38,43] and study commitments [44].

Among middle-aged people, several studies showed that physical activity to a high extent was driven by intrinsic motivation; a willingness to or a desire to do physical activity without any external motivational factors (in contrast to external motivation that may aim at competition results) [35,45,46].

Other studies showed that lack of time was a prominent barrier to physical activity among middle-aged people [36,47,48].

For the elderly, some studies showed that activities that involved social interaction between the participants were found to be motivating [48,49,50]. Well-being (e.g., pleasure) derived from the activity was also mentioned [49,51,52], and one study showed that enjoyment not only facilitated physical activity initiation but also maintenance [48]. Several studies highlighted the expected health benefits as an important motivation for the elderly [33,36,48,49,51,52,53,54,55,56]. Some studies showed that a past history as an active person was a contributing factor to be physically active as an elderly person [50]. Finally, several studies showed that guided activities were motivating for the elderly [49,57]. In addition, a study showed that elderly people who had a more positive attitude towards being active (self-efficacy) were more motivated to remain active [58].

Concerning barriers to physical activity, some studies showed that health-related restrictions, such as having illness or an injury, being afraid of falling (especially in sports-related activities), as well as fatigue and lack of energy were essential barriers for the elderly [33,36,49,57,59,60,61,62].

### 3.2. Socioeconomic Status

A total 21 studies examined socioeconomic status in relation to motives and barriers related to physical activity (Table 3). Some studies found that the motives achievement of health benefits, social interaction and enjoyment were present despite socioeconomic differences [48,49,50,51,52,53,55]. However, some studies found that people with high socioeconomic status were more motivated by well-being, while people with low socioeconomic status were more motivated by health benefits [36,63]. One study showed that a driving factor for participation in physical activity among women of high socioeconomic status was their own positive experiences with physical activity, which led them to have higher levels of self-efficacy [64]. Other studies found that people with low socioeconomic status could be motivated for physical activity through social support and social interaction, including the presence of stigma-free environments [61,65].

One study found that people with low socioeconomic status generally perceived more barriers to physical activity than people with high socioeconomic status [66]. The studies identified several barriers to physical activity among people with low socioeconomic status, including financial restrictions [67], fatigue and lack of energy [43,64,68], health-related restrictions [36,48], shortage of facilities [48,69] and low motivation [64]. A study of women with high and low socioeconomic status, respectively, showed that time restrictions due to work commitments were a barrier among women in the low socioeconomic status group. This group was to a lesser extent capable of planning their everyday lives and thereby creating space for physical activity [64], whereas the high socioeconomic status group had more work flexibility. Conversely, other studies found that time restrictions were a major barrier for people with high socioeconomic status [36,64]. Perceived barriers related to the outdoor environment and shortage of outdoor facilities in the local area were most pronounced in areas with low socioeconomic status groups [70].

### 3.3. Gender

A total 13 studies focused on the gender dimension in relation to motives and barriers related to physical activity (Table 3). Several studies showed that intrinsic motivation was an important indicator of physical activity particularly among women. Self-efficacy—a belief in the ability to maintain participation in regular physical activities—was found to be important, especially for women [35,45,46,71]. Correspondingly, another study showed that social interaction and social support were less important explanatory factors for women’s participation in physical activities than personal factors, such as intrinsic motivation and self-efficacy [64]. However, one study showed that personal (such as self-efficacy), social (such as social support) and environmental factors may jointly explain women’s participation in physical activities [72]. One study found that barriers to physical activity among women were fatigue, time restrictions, lack of energy and low motivation [73]. Furthermore, the study found evidence for a dose-response relationship between the number of barriers identified and meeting recommendations for physical activity among women [73].

Studies on gender differences in motives for physical activity found that men were motivated by competition and physical appearance [67,74]. One study investigated the attitude to change behaviour towards being physically active (readiness to change). The study found that men were more prepared to change their behaviour compared to women [75]. One study focused only on men and found that the primary barriers to physical activity were financial restrictions and lack of knowledge of facilities in their local area [76]. However, other studies found no gender differences in motives and barriers [60,77].

### 3.4. People with Ethnic Minority Background

A total of seven studies dealt with motives and barriers related to physical activity among people with an ethnic minority background (Table 3). Two studies found that sports participation was low among immigrants to Australia from non-English-speaking countries [78,79]. The studies were from the 1990s, and the field of research at that time was sparsely elucidated. However, another 1990s study indicated that migrant participation in major sports such as football and rugby was extensive and believed to be due to migrants’ desire for cultural adaptation and social interaction [79]. A Dutch study of ethnic Dutch and ethnic minority groups (Turks and Moroccans) with no or short education found that social support for physical activity as well as physical activity with others (social interaction) was motivating [65]. The same was true for the experience of health benefits for ethnic minority women (Bosnian, Arab, Filipino and Sudanese) in Australia [80].

According to a survey of female migrants (from Macedonia, Greece, the Netherlands and Poland) in Australia, the most frequent barrier to physical activity was lack of a group to be active with (lack of social interaction), low motivation and time restrictions [78]. Another study examined the barriers to physical activity among asylum seekers from eighteen different countries in Northern England. The study showed that the barriers to physical activity were lack of knowledge of the facilities in the local area as well as the fact that conditions were more luxurious in their new host country (for instance, where they came from, they took the stairs because there were no other alternatives, but in their new host country, there was an elevator instead, which led them to be less motivated to be physically active). Most asylum seekers were motivated to increase their level of physical activity, but competing priorities associated with the asylum seekers’ uncertain future (residence, job situation, etc.) made it difficult to maintain motivation [81]. Religious and cultural dissimilarities were also found to be a barrier to physical activity among Moroccans and Turks living in the Netherlands. For example, men and women may not be able to be active together or at the same place for cultural and religious reasons [81]. Likewise, female immigrants from Southeast Asia may be subject to cultural and religious barriers in relation to coverage of arms, legs and body in connection with physical activity [79]. The study referred to successful programmes where women and men train separately, e.g., swimming only for women [79]. Two studies investigated barriers to physical activity among people with a cultural diversity (African Americans, Latin Americans, and non-Hispanic whites) in the United States. One study showed that time restrictions and priorities, fatigue and cultural norms were barriers to physical activity [43]. The other study showed that both physically active and inactive people indicated the same barriers to physical activity [82].

### 3.5. People with Disabilities

A total of 6 studies focused on motives and barriers related to physical activity among people with disabilities (broadly defined as both physical and mental disabilities) (Table 3). Among the motives for people with disabilities were a belief in the ability to be physically active (‘self-efficacy’) [83], the social interaction of participating in physical activities with others and social support from family and friends [83,84], health benefits and well-being [84], increased sense of bodily capacity [84] as well as the interest in knowing the limits of the body (increased body awareness) [85]. One study found that women with disabilities predominantly chose exercise-oriented rather than sport activities [84].

When it comes to barriers, one study found that people with disabilities perceived more barriers on the personal level (e.g., fatigue, pain and low motivation) [86]. Another study found that time restrictions and experienced disability were the primary barriers [87]. On the societal level, several studies found that public spaces in itself were a barrier to physical activity (e.g., potholes in the streets and lack of access to facilities (bad access conditions)) as well as community norms and stereotypes, etc. (religious and cultural norms) [83,84,86,87]. One study on women with disabilities identified the main barriers to physical activity as a shortage of suitable activities in childhood, few opportunities outside the school setting, lack of interest, a shortage of facilities, a shortage of proper guidance and financial restrictions [88]. Finally, some studies found that people with multiple disabilities and people with more severe disabilities generally experienced more barriers to physical activity [86,87].

An overview of motives and barriers across social backgrounds are presented in Table 4.

## 4. Discussion

This scoping review provides insight into the motives and barriers related to physical activity across social backgrounds including age, socioeconomic status, gender, ethnic minority background and disability status. The analyses identified common motives and barriers but also pointed towards motives and barriers that are specific to or more pronounced among people with certain social background characteristics. The common motives included health benefits, well-being, enjoyment, social interaction, and social support. Common barriers included time restrictions, fatigue and lack of energy, financial restrictions, health-related restrictions, low motivation, and shortage of facilities. However, the review also revealed substantial variations in the most pronounced motives and barriers between people with different social backgrounds. The competition element seems to be particularly important to men. Guided activities seem to be important to elderly people as well as people with disabilities. There seem to be several and grave barriers for both people of low socioeconomic status, people with ethnic minority background and people with disabilities. For people of low socioeconomic status, shortage of facilities in the local community was found to be a prominent barrier. For people with ethnic minority background, religious and cultural norms, lack of knowledge of facilities and lack of social support were found to be central barriers. Among people with disabilities, financial issues, shortage of facilities, bad access conditions to facilities and shortage of proper guidance were identified as central barriers.

Hence, although some types of motives and barriers recur across social backgrounds, the literature review also identified substantial variations in what are the most pronounced motives and barriers between population groups with different social backgrounds. This knowledge may be useful in targeting future health promotion initiatives.

Based on these findings, it is recommendable that efforts to promote physical activity should consider both the common motives and barriers as well as the motives and barriers that were found to be specific to or more pronounced among people with particular social background characteristics.

### 4.1. Implications for Health Promotion

In the Ottawa Charter, WHO defines health promotion as “the process of enabling people to increase control over, and to improve, their health. To reach a state of complete physical, mental, and social well-being, an individual or group must be able to identify and to realize aspirations, to satisfy needs, and to change or cope with the environment” [89].

In this section, we explain how our results may be mobilised in health promotion strategies to improve physical activity and thereby reducing the risk of lifestyle diseases.

To promote better health in the general population, the common motives and barriers identified in this review may be successfully incorporated into overall health promoting initiatives. For instance, social interaction was identified as a common motive for physical activity across social backgrounds, which is a result that could lead to greater focus of health promotion initiatives to strengthen the social aspects of participation in physical activity. Another example of a common motive for physical activity is well-being, which includes, for instance, pleasure with the activity, stress reduction and relaxation. Practitioners who design initiatives to promote physical activity may aim at highlighting these potential benefits of physical activity for mental health when designing health promotion strategies.

To target health promotion initiatives specifically at people with particular social background characteristics, it is recommended to focus on the motives and barriers related to physical activity that are specific to or particularly pronounced among the targeted groups (see results in Table 4 for inspiration). Ideally, all items in the table should be considered, when practitioners seek to promote physical activity among a group of people with similar social background characteristics. When promoting a physical activity for young adults, the activity should, for instance, be fun, and it should be easy to access the facilities. Likewise, when promoting a physical activity for people with disabilities, the activity should target positive experiences of bodily capacity and provide suitable facilities and guidance. Further, for men, elements of competition may be successfully incorporated in interventions to promote physical activity.

Policy makers can also learn from the findings of this scoping review when seeking to reduce social inequalities in physical activity and health in the sense that social background must be considered when designing policies. Particularly, the motives and barriers of social groups that are less physically active than other groups (e.g., people with low socioeconomic status, people with disabilities and people with ethnic minority background) should be given particular attention in the policy design phase.

### 4.2. Strengths and Limitations

One strength of this scoping review was that the analysis permitted us to divide motives and barriers across social backgrounds and particularly to point out specific motives and barriers for each population group. Furthermore, the literature search method was extensive and systematic, and it has provided a broad insight into the literature in this field. It was a strength that we searched in four scientific databases, which were all found to provide relevant articles relating to the purpose of the review.

However, this review also has limitations. Because most of the studies included in our scoping review were cross-sectional and because some findings are reported based on only on or a few studies, the results should generally be interpreted with some caution. Especially where causal relationships are postulated.

By focusing on five elements of social background, namely age, socioeconomic status, gender, ethnic minority background and disability status, the review does not include studies on other potentially relevant social groups (e.g., people with a chronic disease). Moreover, studies of population groups that had a limited transferability to Danish conditions (e.g., African Americans) and groups with a limited prevalence (e.g., people with a rare disability) were excluded.

As this was a scoping review, a formal synthesis of evidence was not undertaken. However, the goal of a scoping review is to identify the research, which is conducted, to provide a descriptive summary and not necessarily to assess the quality [27,90].

Finally, as the results were based on a majority of studies from European countries, Australia and Canada, the results may not be representative for third-world countries or North America.

### 4.3. Implications for Future Research

This review uncovers a lack of longitudinal studies to investigate the causal relationships between motives and barriers related to physical activity across social backgrounds as the vast majority of the studies included in the review were cross-sectional. However, this review provides a foundation to build future intervention study designs.

Furthermore, the exclusion of studies that focused either on people with disabilities with a limited prevalence or people from ethnic minority groups not relevant in a Danish context leaves potential for literature reviews on motives and barriers particularly among these groups. It is, therefore, recommended to undertake more focused systematic reviews with focus on the motives for and barriers to physical activity for people with an ethnic minority background and people with disabilities.

## 5. Conclusions

The present study provides insight into the motives and barriers related to physical activity across social backgrounds including age, socioeconomic status, gender, ethnic minority background and disability status. The analyses show both common motives and barriers but also point towards motives and barriers that are specific to or more pronounced among people with certain social background characteristics.

Based on these findings, it is recommendable that efforts to promote physical activity should consider both the common motives and barriers as well as the motives and barriers that were found to be specific to, or more pronounced among, people with particular social background characteristics. This knowledge may be useful in targeting future health promotion initiatives.

## Figures and Tables

**Figure 1 ijerph-18-05810-f001:**
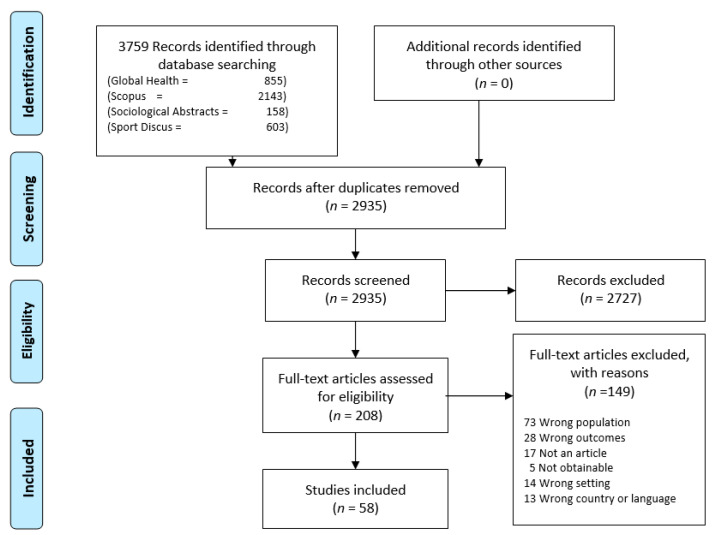
Flowchart of search results.

**Table 1 ijerph-18-05810-t001:** Search terms in the four selected databases.

	Searchblocks	Physical Activity	Motivation	Social Background
Database				
**Scopus**		Free text words: Sport * W/2 participati * Physical W/2 exercis * Physical W/2 activit *	Free text words: Motivat * Unmotivat * Demotivat * Amotivat * Barrier * Self W/2 determination W/2 theor * Social W/2 cognitive W/2 theor *	Free text words: Socioeconomic W/2 status Social W/2 class * Vulnerable W/2 group * Education * W/2 level * Education * W/2 background Ethnic W/2 minorit * Handicap * Disabilit * Disabled
**SPORTDiscus**		Defined keywords: DE “SPORTS participation” DE “SEDENTARY behaviour” DE “SEDENTARY lifestyles” DE “SEDENTARY people”	Defined keywords: DE “MOTIVATION (psychology)” DE “TRANSTHEORETICAL model of change”	Defined keywords: DE “MINORITIES in sports” DE “PEOPLE with disabilities”
		Free text words: (Sport * N2 participati *) (Physical N2 exercis *) (Active N2 living) (Active N2 transportati *) (Physical N2 inactivity)	Free text words: Motivat * Barrier * constraint * (Self N2 determination N2 theor *) (Social N2 cognitive N2 theor *) (Readiness N2 change *)	Free text words: (Socio N2 economic N2 status) (Social N2 class *) (Vulnerable N2 group *) (Education * N2 level *) (Education * N2 background) (Ethnic N2 minorit *) Handicap * Disabilit * Disabled
**Global Health**		Defined keywords: DE “sport” DE “physical activity” DE “active recreation	Defined keywords: DE “motivation”	Defined keywords: DE “socioeconomic status” DE “social classes” DE “social status” DE “ethnic groups” DE “people with disabilities”
		Free text words: Same as SPORTDiscus	Free text words: Same as SPORTDiscus	Free text words: Same as SPORTDiscus
**Sociological Abstracts**		Defined keywords: MAINSUBJECT.EXACT (“Sports Participation”)	Defined keywords: MAINSUBJECT.EXACT.EXPLODE (“Motivation”) MAINSUBJECT.EXACT.EXPLODE (“Constraints”)	Defined keywords: MAINSUBJECT.EXACT.EXPLODE (“Socioeconomic Status”) MAINSUBJECT.EXACT.EXPLODE (“Social Class”) MAINSUBJECT.EXACT.EXPLODE (“Social Status”) MAINSUBJECT.EXACT.EXPLODE (“Handicapped”)
		Free text words: Sport * NEAR/2 participati *) (Physical NEAR/2 exercis *) (Active NEAR/2 living) (Active NEAR/2 transportati *) (Physical NEAR/2 inactivity)	Free text words: Motivat Barrier * constraint * (Self NEAR/2 determination NEAR/2 theor *) (Social NEAR/2 cognitive NEAR/2 theor *)	Free text words: Socio NEAR/2 economic NEAR/2 status) (Social NEAR/2 class *) (Vulnerable NEAR/2 group *) (Education * NEAR/2 level *) (Education * NEAR/2 background) (Ethnic NEAR/2 minorit *) Handicap * Disabilit * Disabled

* (Truncations) include as many forms of the search words as possible.

**Table 2 ijerph-18-05810-t002:** Descriptive information about the origin (continent and country) of the included studies.

Europe	No. of Studies
Albania	1
Austria	1
Belgium	3
Denmark	1
Estonia	1
Finland	1
France	1
Germany	3
Greece	3
Norway	1
Spain	1
Switzerland	1
Netherlands	1
United Kingdom	7
**Oceania**	
Australia	14
**North America**	
Canada	1
USA	11
**More than one country**	6

Bold format: Continent and studies with more than one country.

**Table 3 ijerph-18-05810-t003:** Number of studies structured according to their main foci.

Social Background	Studies
Age	31
Socioeconomic status	21
Gender	13
People with ethnic minority background	7
People with disabilities	6

**Table 4 ijerph-18-05810-t004:** Motives and barriers related to physical activity across social backgrounds.

			Motives and Barriers Related to Physical Activity	
			Motives or Motivation	Barriers
**Social Background**	Age	Young adults (16–40) years)	Enjoyment Physical appearance Social interaction	Shortage of facilities Shortage of suitable activities Time restrictions
		Middle-aged people (40–60 years)	Intrinsic motivation	Time restrictions
		Elderly people (60+ years)	Enjoyment Guided activities Health benefits Self-efficacy Social interaction Well-being	Fatigue and lack of energy Health-related restrictions
	Socioeconomic status	High socioeconomic status	Enjoyment Health benefits Self-efficacy Social interaction Well-being	Time restrictions
		Low socioeconomic status	Enjoyment Health benefits Social interaction Social support	Fatigue and lack of energy Financial restrictions Health-related restrictions Low motivation Shortage of facilities Time restrictions
	Gender	Women	Environments Intrinsic motivation Self-efficacy Social support	Fatigue and lack of energy Low motivation Time restrictions
		Men	Competition Physical appearance Readiness to change	Financial restrictions Lack of knowledge of facilities
	People with ethnic minority background		Health benefits Social interaction Social support	Fatigue and lack of energy Lack of knowledge of facilities Lack of social interaction Low motivation Religious and cultural norms Time restrictions Uncertain future
	People with disabilities		Increased body awareness Self-efficacy Social interaction Social support Well-being	Bad access conditions Fatigue and lack of energy Financial restrictions Low motivation Religious and cultural norms Shortage of facilities Shortage of proper guidance Time restrictions

## Data Availability

All data has been published as Appendix A and was collected from published articles.

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
