# Peer review of "Motives and Barriers Related to Physical Activity and Sport across Social Backgrounds: Implications for Health Promotion"

_ijerph, 2021, doi:10.3390/ijerph18115810_

Round 1

Reviewer 1 Report

Dear authors, 

The study is well written, and the authors place the GAP well and highlight the contributions of their research over previous research. However, I am a little confused about the included and excluded papers in this review. Some suggestions as follows:

  1. There are vast scientific evidences from different countries showing that built environment is an important factor affecting physical activity. Also, some of previous studies investigate different social backgrounds. For example, Smith et al. found that older females used parks and greenspaces at a lower frequency than older males did, and parks and greenspaces only promoted older people’s leisure-time physical activity in males [1]. Yu et al. found that older people’s leisure-time physical activity was affected by various built environment elements in different sex groups, and older females’ leisure-time physical activity level was more sensitive to the built environment [2]. However, built environment was little discussed in this paper and little mentioned in the results table 4. The authors please consider whether the relative papers investigate the association of built environment with physical activity should be included in this review.

[1] Smith, M.; Hosking, J.;Woodward, A.;Witten, K.; MacMillan, A.; Field, A.; Baas, P.; Mackie, H. Systematic literature review of built environment effects on physical activity and active transport—An update and new findings on health equity. Int. J. Behav. Nutr. Phy. 2017, 14.

[2] Yu, J.; Yang, C.; Zhang, S.; Zhai, D.; Wang, A.; Li, J. The Effect of the Built Environment on Older Men’s and Women’s Leisure-Time Physical Activity in the Mid-Scale City of Jinhua, China. Int. J. Environ. Res. Public Health 2021, 18, 1039.

  1. Most of included studies are from European countries, Australia and Canada. To our knowledge, quite a few papers from other countries also investigated the factors affecting physical activity, such as paper from China, also from other Asian countries. However, None of Asian countries studies were included in the review. Why those papers were excluded from this review?

  1. Some papers matching the purpose of this review were not included in this study, such as the two below papers, and not limited in the two papers. Please consider whether the include and excluded rule of this study were correct or could be modified.

[1] Spiteri, K.; Broom, D.; Bekhet, A.H.; de Caro, J.X.; Laventure, B.; Grafton, K. Barriers and Motivators of Physical Activity Participation in Middle-Aged and Older Adults-A Systematic Review. J. Aging Phys. Activ. 2019, 27, 929–944.

[2] Baert, V.; Gorus, E.; Mets, T.; Geerts, C.; Bautmans, I. Motivators and barriers for physical activity in the oldest old: A systematic review. Aging Res. Rev. 2011, 10, 464–474.

  1. table 2 and table 3 are better in the three-line table form.

Reviewer 2 Report

The authors used a scoping review in order to answer the question whether motives and barriers related to regular physical activity are different across different social backgrounds. This is an important issue because WHO emphasizes that for a successful implementation of the physical activity guidelines the needs of different target groups have to be considered. The paper is well written and contains informative tables.

There are two minor comments:

Materials and methods, page 2, line 85: How did you define that a search result was relevant or not? Please clarify this.

Results, page 22, paragraph 22 studies examined. Several times it says: One study showed… As we know, one study is not enough to say something is “true” or not. In addition, most of the studies are cross-sectional studies. From my point of view, it is important to mention these two limitations and the consequences of these limitations regarding the results of your study.  

Reviewer 3 Report

IJERPH – 1208420

Reviewer Report: Motives and barriers related to physical activity and sport across social backgrounds: implications for health promotion

Many thanks for the invitation to review this manuscript. The manuscript describes a scoping review to examine motives and barriers related to physical activity and sport among people with different social backgrounds. I do not feel as if the paper makes a clear or strong enough rationale for focusing on this area of study given the volume of research which is already available on this topic. My comments below will hopefully serve to add clarity and strengthen the paper. Consequently, I recommend that this paper be subject to major revisions before it is considered for publication.

Introduction

There is much review level evidence available which has examined barriers and facilitators to individuals participation and the correlates of physical activity as well as research demonstrating the importance of social background (e.g Socio-cultural determinants of physical activity across the life course: A 'Determinants of Diet and Physical Activity' (DEDIPAC) umbrella systematic literature review 10.1186/s12966-017-0627-3) however little reference has been made to this work to allow the reader to understand why there is a need for this particular review, what niche it fills and how this review fits within the existing literature. In addition, given that, it is also unclear why a scoping review was chosen as opposed to a systematic review for example?

Line 30 – I would question weather this is the most appropriate reference given the volume of high quality epidemiological data available?  I’d combine line 30 and 31 so that the Lancet reference (Ref2) can be used.

Line 61-63 repetition of lines 58 and 59

Line 76 I think the term ‘participation’ would be better than ‘sporting behaviour’ as this generally refers to gamesmanship/sportsmanship

Materials and methods

Can the authors comment on why different search terms were used in different databases and what effect this is likely to have had on the findings.

I feel as if the in-and exclusion criteria would benefit from being described ina  little ore detail, at the moment these appear quite broad

Line 131 – I am unclear of the significance of a Danish cntext referred to hear, this hasn’t previously been mentioned

I think the methods section would benefit from more information in terms of analysis and how the extracted data was subsequently synthesised. For example was the data coded into apriori categories/themes?

Results

Can the authors provide additional information/justification for the use of the age groupings? I am unsure 16-40 years would be considered as ‘young people’ as there is large variation within this group and motives and barriers change over time. The United Nations definition of youth for example is 15-24 years.

Table 4 is a very useful summary table, for clarity I would list the motives and barriers in alphabetical order for each row.

Discussion

Line 351 Really pleased there is reference to implications for health promotion included within the paper however these seem very surface level suggestions. It would also be useful to include implications for policy makers in terms of reducing inequalities.

Line 400 I am unsure of the usefulness of this statement as RCTs are not the best design to examine the relationship between motives and barriers to PA when it is so contextually 

Round 2

Reviewer 1 Report

Dear authors,

Thank you for the improvements of this paper. my questions have been answered. 

Reviewer 3 Report

Thank you for the revised version I am happy that the changes have improved the manuscript